# Dimerization regulates both deaminase-dependent and deaminase-independent HIV-1 restriction by APOBEC3G

Michael Morse[1], Ran Huo[1], Yuqing Feng[2], Ioulia Rouzina[3], Linda Chelico[2] & Mark C. Williams [1]

APOBEC3G (A3G) is a human enzyme that inhibits human immunodeficiency virus type 1 (HIV-1) infectivity, in the absence of the viral infectivity factor Vif, through deoxycytidine deamination and a deamination-independent mechanism. A3G converts from a fast to a slow binding state through oligomerization, which suggests that large A3G oligomers could block HIV-1 reverse transcriptase-mediated DNA synthesis, thereby inhibiting HIV-1 replication. However, it is unclear how the small number of A3G molecules found in the virus could form large oligomers. Here we measure the single-stranded DNA binding and oligomerization kinetics of wild-type and oligomerization-deficient A3G, and find that A3G first transiently binds DNA as a monomer. Subsequently, A3G forms N-terminal domain-mediated dimers, whose dissociation from DNA is reduced and their deaminase activity inhibited. Overall, our results suggest that the A3G molecules packaged in the virion first deaminate viral DNA as monomers before dimerizing to form multiple enzymatically deficient roadblocks that may inhibit reverse transcription.

[1] Department of Physics, Northeastern University, 360 Huntington Avenue, Boston, MA 02115, USA. [2] Department of Microbiology and Immunology, University of Saskatchewan, 107 Wiggins Road, Saskatoon, SK, Canada  S7N 5E5. [3] Department of Chemistry and Biochemistry, Ohio State University, 100 West 18th Avenue, Columbus, OH 43210, USA. Correspondence and requests for materials should be addressed to M.C.W. (email: mark@neu.edu)

The APOBEC3 family of proteins are a set of enzymes with antiviral properties that are found in humans and primates. APOBEC3G (A3G) is particularly effective at suppressing the infectivity of human immunodeficiency virus type 1 (HIV-1)[1, 2]. As a result, HIV-1 must circumvent this protection through the expression of viral infectivity factor (Vif)[3–5], which is able to target A3G for degradation[6, 7]. If A3G escapes Vif-mediated degradation it is able to become encapsidated into newly forming virions through its association with genomic RNAs that are bound by HIV-1 Gag. Once the virion infects a new host cell, the RNA genome will be reverse transcribed into the (−)DNA and the RNA degraded. During this synthesis step and before the (+)DNA is formed, encapsidated A3G, a single-stranded DNA (ssDNA) deoxycytidine deaminase[8, 9], catalyzes the deamination of deoxycytidine to form the promutagenic deoxyuridine on single-stranded (−)DNA. The subsequent use of these deoxyuridines as a template for (+)DNA synthesis results in C-G to T-A mutations in the proviral DNA and impairs viral replication[10–12]. It has been previously shown, however, that non-catalytic A3G mutants retain some anti-viral functionality[13–17], indicating the presence of a deamination-independent mode of HIV-1 protection[18–20].

The processivity of an A3G enzyme is a determinant of HIV-1 restriction efficiency. A3G must catalyze the deamination of viral cytosines on (−)DNA during the synthesis of the proviral DNA. This dynamic process exposes the A3 substrate, viral ssDNA, for only a limited amount of time. In order to search for and locate deamination motifs efficiently to induce a large number of mutations, A3G uses a combined one-dimensional and three-dimensional diffusional search mechanism involving movements termed sliding, jumping and intersegmental transfer[21–23]. These movements are collectively termed facilitated diffusion and are similar to the search mechanisms described for the lac repressor and restriction enzymes[24, 25], except that A3G searches ssDNA, rather than double-stranded DNA (dsDNA). The enzyme uses short-range sliding movements (≤20 nt) by sliding along the DNA phosphate backbone to search for deamination motifs in a local area. To ensure an efficient search, A3G also has a long-range movement termed jumping that enables the enzyme to move large distances from the previous search site by diffusing in the charged domain of the ssDNA, without direct binding. As a result, a large number of nucleotides can be searched with a combination of these two mechanisms. In addition, if the synthesis of proviral DNA by reverse transcriptase could be temporarily stalled or impeded, the additional time could allow for further deamination of the (−)DNA[26, 27].

Structurally, A3G has two cytidine deaminase domains, the properties of which control A3G function. The N-terminal domain (NTD) is catalytically inactive, whereas the C-terminal domain (CTD) is catalytically active and responsible for deaminase activity[28, 29]. The NTD is important, however, in the binding of A3G to RNA and DNA, processivity of deamination activity, as well as in incorporation of A3G into virions[22, 30, 31]. Vif also interacts with A3G through the NTD[7, 32]. In addition to interactions with nucleic acids, individual A3G monomers can associate with one another to form ordered dimers, tetramers and larger oligomeric structures, which have been directly observed using both atomic force microscopy (AFM) imaging[33–35] and size-exclusion chromatography[30, 36]. The exact role of oligomerization in A3G function, however, is not completely understood. Some studies have observed A3G in solution to be in a dimeric form before binding nucleic acids[30, 36], whereas others have observed A3G in a primarily monomeric form[30, 33, 35]. It has been shown that the highly basic NTD is the primary dimerization interface for A3G bound to nucleic acids[30, 31, 37, 38],

although dimerization centered at the CTD has been observed as well[39]. In addition, for the formation of tetramers and larger structures, both domains must be able to self-associate. How the various oligomerization states of A3G affect its biological functions is a continued topic of debate, although it has been shown that mutations of the NTD designed to disrupt oligomerization on RNA also prevent A3G from being packaged in the virion[31]. There is also no clear consensus on how oligomerization impacts A3G's ability to deaminate viral ssDNA. A3G's preference to form dimers[33] or larger oligomeric structures[35, 36, 40] when bound to ssDNA has raised the question of whether oligomerization is required for catalytic activity. In addition, one study observed that conditions that promote the formation of large A3G oligomers are also conducive to deaminase activity[41]. However, oligomerization-deficient A3G mutants, which remain primarily in a monomeric state, are still able to effectively deaminate ssDNA[30, 31, 42], although the deaminase activity of these A3G mutants is less processive and less able to bypass dsDNA regions via jumping than wild-type (WT) A3G[22, 30]. Thus, it is possible that dimerization may impact enzymatic processivity, or alternatively, that the critical amino acids that promote oligomerization also control processivity.

One way to determine the effect of oligomerization on A3G function is to examine the behavior of various oligomerization-deficient mutants. Structures of both domains have been obtained[38, 43, 44], which can be used to determine the residues critical to forming the shared interface between A3G monomers when dimers and tetramers are formed. In particular, the hydrophobic residues Phe126-Trp127 of the NTD are believed to form a hydrophobic core at the dimer interface, based on the structure of the homologous APOBEC2 tetramer[45]. More recent crystal structures of the rhesus macaque A3G NTD have confirmed this result[38]. In the CTD, the residues Ile314-Tyr315 and Arg313-Asp316-Asp317-Gln318 may also form a dimer interface[43]. As a result, the NTD mutant F126A-W127A (FW) and two CTD mutants R313A-D316A-D317A-Q318A (RDDQ) and I314A-Y315A (IY) have been previously developed and shown to reduce the degree of A3G oligomerization[30]. A structural model of A3G dimerized in a 'head to tail' orientation (the NTD of one monomer interacting with the CTD of a second monomer) has also suggested that some of these residues may be critical to A3G oligomerization and function[46].

The oligomerization of A3G may also be critical to the deamination-independent mode of HIV-1 inhibition. According to the previously proposed roadblock model, HIV-1 replication is inhibited by the presence of A3G bound to viral minus strand ssDNA, physically preventing reverse transcription (RT)[20]. Such a model would require slow, stable binding of A3G to ssDNA, which would interfere with the enzyme's ability to quickly bind, deaminate, and unbind up to 1000 sites along the minus strand DNA template[9]. We have recently shown, however, that A3G is able to exhibit both fast and slow binding kinetics, which can be minimally described by a two state binding model in which A3G subunits first bind ssDNA in a 'fast' binding state and then later form stably bound oligomers in a 'slow' binding state[47].

$$\text{A3G} + \text{ssDNA} \underset{k_{-1}}{\overset{ck_1}{\rightleftharpoons}} (\text{A3G} + \text{ssDNA})_{\text{fast}} \underset{k_{-2}}{\overset{k_2}{\rightleftharpoons}} (\text{A3G} + \text{ssDNA})_{\text{slow}}$$

(1)

Such a system would allow A3G to first transiently bind and quickly deaminate the viral ssDNA and then oligomerize, forming a roadblock for RT, while bound for an extended period of time. However, the degree of oligomerization (dimerization,

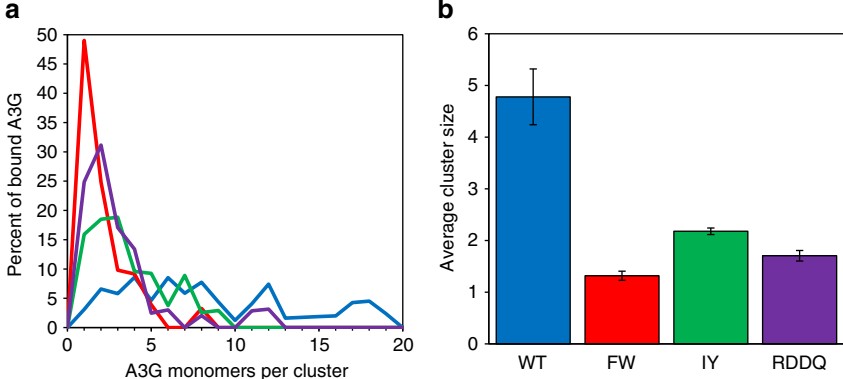

**Fig. 1** Measurements of A3G oligomerization using AFM imaging. **a** The number of monomers per A3G cluster formed on ssDNA are plotted in a histogram. FW A3G (NTD mutant) is mostly bound to the ssDNA in a monomeric form. For IY and RDDQ A3G (CTD mutants) the most common state is a dimer. WT A3G is more likely to form tetramers and larger oligomers as compared to the mutants. **b** Comparison of average cluster size for WT and mutant A3G. Mutating the CTD moderately reduces oligomerization whereas mutating the NTD disrupts oligomerization to an even greater degree. *Error bars* are SE based on the size of individual protein clusters formed on at least seven DNA constructs per A3G variant

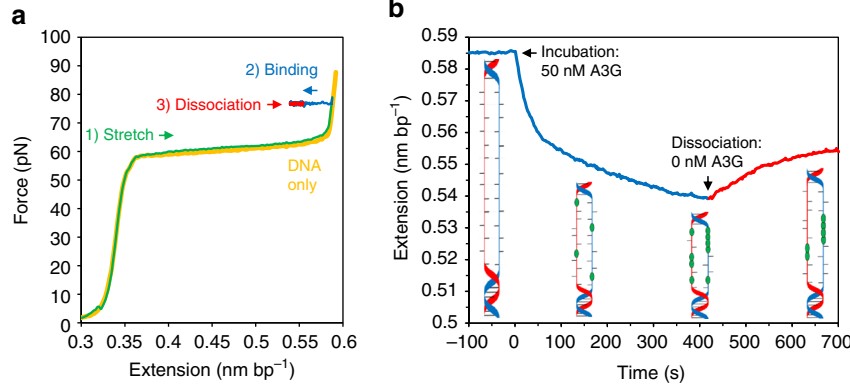

**Fig. 2** Measuring A3G binding to ssDNA using optical tweezers. **a** A single dsDNA molecule is stretched until reaching a tension of 80pN (1). Above ~60 pN, a force induced melting transition creates two ssDNA strands. A3G is flowed into the sample allowing for binding to ssDNA, (2) whereas active force feedback maintains constant force. After a set incubation time, buffer without A3G is flowed into the sample, resulting in unbinding of some of the bound A3G (3). **b** A3G begins to bind ssDNA, resulting in a shortening of the DNA extension at time 0 s (*blue*). At time 400 s, buffer replaces the A3G and some fraction of the A3G dissociates, resulting in an increase in extension (*red*)

tetramerization, etc.) that is needed for A3G to transition from the initial binding state to the more stable second state is not known. This is of particular importance, because this model would also imply that sufficiently large oligomers lose their catalytic efficiency due to decreased substrate cycling, which is controversial, as discussed above.

To understand how oligomerization may regulate A3G activity, we use an optical tweezers apparatus with high spatial and force resolution and a fast response rate to directly measure both bimolecular A3G-ssDNA binding and oligomer formation at the single molecule level. We observe these kinetics for WT A3G, as well as NTD and CTD mutants, in order to determine the role of each domain in A3G binding and oligomerization. Our results also suggest that, in addition to significantly increasing the duration of a binding event on ssDNA, oligomerization decreases A3G's enzymatic activity. We also show that dimerization alone, rather than the formation of large oligomers, is sufficient to dramatically slow A3G dissociation from DNA and reduce A3G deamination efficiency.

## Results

**A3G dimerization mutations strongly reduce oligomerization.** We utilized AFM imaging to confirm that mutating the NTD and

CTD as described above reduces the degree of oligomerization exhibited by A3G. Linearized ssDNA was incubated with either WT A3G or mutant A3G and imaged in liquid on a coated mica surface (Supplementary Fig. 1). A height threshold was applied to identify A3G subunits bound to the ssDNA. The bound A3G form clusters in which multiple subunits are bound adjacent to each other as a result of oligomerization. We determined the total number of A3G subunits in each cluster based on its total volume in relation to that of a single A3G monomer. The results show that, when WT A3G binds ssDNA, the majority of subunits form dimers, tetramers and other larger oligomers. When the NTD is mutated (FW), A3G's monomeric form is the most frequently observed, whereas larger oligomers are mostly absent (Fig. 1). In contrast, for the CTD mutants (RDDQ and IY), dimers are most frequently observed and larger oligomers are present in reduced numbers. These results are consistent with the NTD being the primary dimerization interface and with the CTD acting as a secondary interface that allows for the formation of tetramers and larger oligomers. In summary, when bound to ssDNA, we observe that the NTD mutant A3G (FW) remains primarily in a monomeric state, the CTD mutants (IY and RDDQ) form dimers mediated by the NTDs but are deficient in forming larger

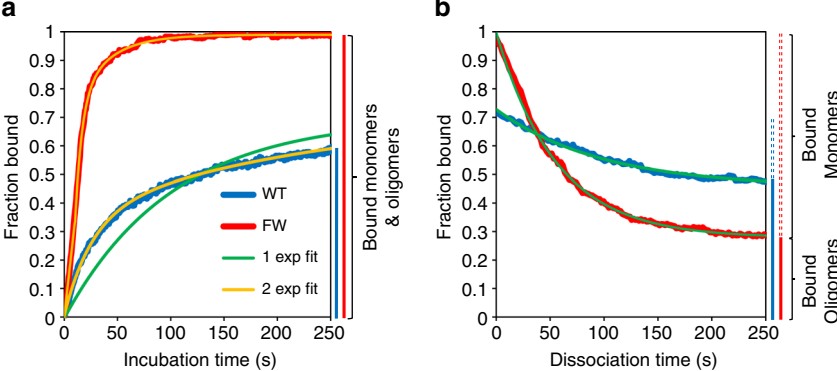

**Fig. 3** Fraction of ssDNA bound by A3G over time during incubation and dissociation. **a** During incubation (at 50 nM in sample data), the amount of A3G bound increases monotonically before asymptotically reaching an equilibrium value. As compared with the binding of FW mutant, binding of WT A3G does not reach full saturation even at long incubation times and high A3G concentrations. The binding kinetics cannot be represented as a single exponential function consistent with a simple one step binding system. Instead, two binding rates are used to fit the binding curve ($k_{fast}$ and $k_{slow}$), as predicted by the two step binding model. **b** During dissociation in buffer (after 200 s incubation time in sample data), only a fraction of the bound A3G unbinds from the ssDNA. More A3G remains bound for WT A3G as compared with FW mutant. Only one rate ($k_{-1}$) is needed to fit the dissociation curves for both WT and mutant A3G. After equal incubation times, a smaller fraction of FW mutant A3G has formed oligomers as compared with WT

oligomeric structures and WT both dimerizes through the NTD and forms larger oligomers through the CTD.

**A3G binding kinetics directly measured by optical tweezers.**
A single double-stranded λ-DNA molecule was tethered between two polystyrene beads in a flow cell. One bead was held by a micropipette tip affixed to a piezoelectric stage, which can be moved with single nanometer resolution. The other bead was held in a stationary, dual-beam optical trap. By moving the pipette tip, the DNA can be stretched to a specific length, resulting in tension along the DNA based on its well-characterized mechanical polymer properties. This force exerted on the DNA can be determined with sub-pN resolution based on the measured deflections of the trapping beam. This apparatus allows for the simultaneous measurement of both DNA extension and tension on a 10 ms timescale. Under our experimental conditions, exerting a force of over 60 pN on the DNA will result in force-induced melting, converting the DNA construct into ssDNA form[48, 49]. Thus, in order to measure the binding of A3G to ssDNA specifically, the construct was first stretched until a force of 80 pN was achieved (Fig. 2). A solution containing a known concentration of WT or mutant A3G was then flowed into the sample chamber. A3G binding to ssDNA results in a small characteristic reduction in extended length. Previous studies have suggested A3G wraps ssDNA during binding[33, 50], which would account for this effect. This contraction of ssDNA due to A3G binding has also been previously observed using FRET measurements[51]. Thus, while a constant force of 80 pN is maintained by the instrument, we observe a decrease in DNA extension proportional to the amount of A3G bound. After a set incubation time, the A3G was removed from the system by flowing in a buffer solution. Following buffer flow, the DNA extension is observed to slightly increase as some fraction of the bound A3G dissociates. However, the initial DNA extension is not fully recovered, indicating that some A3G remains bound in a slower dissociating state. The ssDNA extension as a function of time during both binding and dissociation (Fig. 2b) were extracted for analysis. This was repeated at different A3G concentrations (Supplementary Fig. 2) and with the A3G mutants (Supplementary Fig. 3). Previously, the binding of WT A3G to ssDNA was measured using a similar technique in which incubation was allowed for set times at high force before allowing

for partial reannealing of dsDNA at lower forces[47]. The total binding at discrete times was inferred based on the number of base pairs able to reanneal and these data points were fit to the two state binding model discussed above. Now, however, we are able to directly observe both binding and dissociation of A3G in real time at a constant force without interference with dsDNA base pairing, allowing for more precise measurements of binding kinetics. Although the application of high forces could reduce A3G binding as such binding shortens ssDNA, we observed strong binding to stretched ssDNA. This is likely to be due to the fact that the change in ssDNA length due to A3G binding, though measurable, is relatively small, with a maximum value that is < 10% of the total ssDNA contour length at that force.

**Two-step binding and oligomerization model for A3G activity.**
To determine the total fraction of ssDNA bound by A3G at a given time, we assume that the reduction in ssDNA extension due to binding is directly proportional to the number of bound A3G subunits. Before A3G binding, ssDNA has an extended length of $0.586 \pm 0.001$ nm bp$^{-1}$ when stretched to 80 pN. At saturated binding, the length of ssDNA is reduced by $0.051 \pm 0.003$ nm bp$^{-1}$. Using these parameters, we calculate the fractional binding of A3G as a function of time. During incubation, the amount of A3G bound increases over time and eventually saturates. The time dependence of bound ssDNA does not fit to a single-exponential function (Fig. 3a). To adequately fit the data, the binding curves are fit to a function with two exponential rates, consistent with a two-step binding model (Eq. 1), as was previously observed for WT A3G[47]. However, now that we can observe the binding of A3G to ssDNA in real time, we measure both the rates of A3G-ssDNA binding and A3G oligomerization directly:

$$f_{incubation}(t) = A_{fast}(1 - e^{-k_{fast}t}) + A_{slow}(1 - e^{-k_{slow}t}) \quad (2)$$

Here $A_{fast}$ and $A_{slow}$ are the fractions of the ssDNA strand that are initially bound by A3G before oligomerization and that eventually become bound due to oligomerization, respectively. In this model, the first step is a bimolecular process (free A3G binding to ssDNA), whereas the second step is unimolecular (bound A3G-forming oligomers). Thus, the first, fast step is linearly dependent on the concentration of A3G and the second,

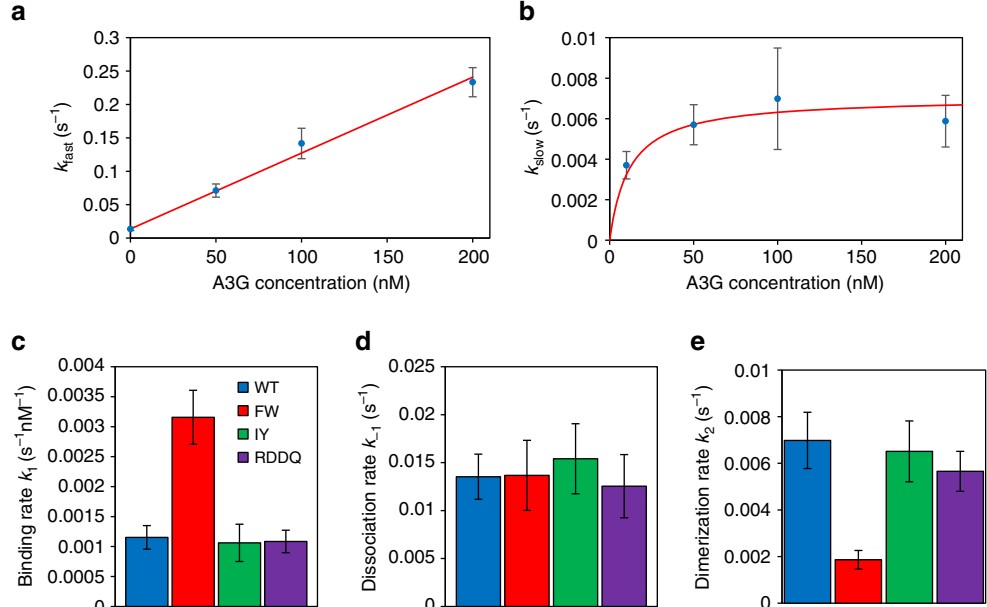

**Fig. 4** Measured rates of binding and oligomerization of A3G on ssDNA. **a**, **b** fast and slow binding rates for WT as a function of concentration. The data point 0 nM is the average single exponential rate fit to dissociation curves (Fig. 3b). All other data points are the average fast (**a**) and slow (**b**) exponential rates fit to incubation curves at varying WT A3G concentrations. The expected behavior of $k_{fast}$ and $k_{slow}$ are plotted (*red lines*) based of the derived values of $k_1$, $k_{-1}$ and $k_2$. $k_{fast}$ is linear with concentration and $k_{slow}$ saturates at high concentrations ($ck_1 \gg k_{-1}$). **c–e** Concentration independent, bimolecular binding rate $k_1$, unbinding rate $k_{-1}$, and rate of oligomerization $k_2$ for WT A3G and mutants. The NTD mutant (FW) binds to ssDNA at an increased rate but forms oligomers much more slowly than the WT and CTD mutants. All error bars are SEM based on at least three independent experiments and resulting fitting parameters per condition

slow step is dependent on the equilibrium amount of A3G bound. The fast and slow binding terms can be written in terms of the concentration-dependent binding rate $ck_1$, dissociation rate $k_{-1}$, rate of oligomer formation $k_2$, and rate of oligomer disassembly $k_{-2}$:

$$k_{fast} = ck_1 + k_{-1} \tag{3}$$

$$k_{slow} = \left(\frac{ck_1}{ck_1 + k_{-1}}\right)k_2 + k_{-2} \tag{4}$$

In addition to direct measurements of A3G association to ssDNA, we also measured the dissociation of A3G from ssDNA in buffer only after incubation times ranging from 200 to 1000 s. In this case, the data could be fit well with a single exponential dependence on time.

$$f_{dissociation}(t) = A_{total} - A_{dis}\left(1 - e^{-k_{-1}t}\right) \tag{5}$$

Here $A_{total}$ is simply the total amount of protein bound after the set incubation time and $A_{dis}$ is the amount of A3G that unbinds from the ssDNA, which occurs at rate $k_{-1}$. Further unbinding of A3G may occur at much longer timescales such that all bound protein will dissociate given enough time. However, only a single rate of unbinding is observed under these conditions and times of observations (Fig. 3b). Both the measured amplitudes of the processes ($A_{fast}$, $A_{slow}$, $A_{total}$ and $A_{dis}$) as well as the rates at which they occur ($k_{fast}$ and $k_{slow}$) reveal important information about A3G activity.

Using the $k_{fast}$ and $k_{slow}$ fitting parameters obtained from multiple binding experiments in combination with Eqs. 3 and 4, we extract the elementary rates of A3G binding ($k_1$), dissociation ($k_{-1}$) and oligomerization ($k_2$). First, we have directly measured $k_{-1}$ as the sole exponential rate in our dissociation curves (Fig. 3a). Based on the average results of 16 independent measurements, we find WT A3G dissociates at a rate $k_{-1} =$

$0.014 \pm 0.002 \, s^{-1}$. Next, we plug this measured $k_{-1}$ rate, along with the known A3G concentration $c$ and $k_{fast}$ parameter from every binding curve into Eq. 3 and solve for the value of $k_1$. We find an average WT A3G association rate constant of $k_1 = 0.0012 \pm 0.0002 \, nM^{-1} s^{-1}$. Finally, we use both the rates $k_{-1}$ and $k_1$ along with the $k_{slow}$ parameter from every experiment to obtain the value of $k_2$ from Eq. 4. It is noteworthy that $k_{-2}$ is negligibly small on the time scale of these experiments. We were unable to remove most bound A3G from the ssDNA substrate even when washing in buffer for up to 100 min. Thus, under saturating conditions $k_2$ itself is approximately equal to $k_{slow}$. WT A3G oligomerizes at an average rate $k_2 = 0.0070 \pm 0.0012 \, s^{-1}$ at saturating binding conditions. As a check of the model, we also plot the average value of the $k_{fast}$ and $k_{slow}$ fitting parameters as a function of A3G concentration along with their expected concentration dependence from Eqs. 3 and 4 (Figs. 4a, b). It is noteworthy that at $c = 0$, $k_{fast}$ is simply the dissociation rate $k_{-1}$. As expected, $k_{fast}$ varies linearly with A3G concentration, whereas $k_{slow}$ does not vary significantly over the range of A3G values we are investigating. When $ck_1 \gg k_{-1}$, A3G binding will saturate the ssDNA substrate to maximize interactions between bound A3G monomers, which in turn maximizes the rate of oligomer formation. Agreement between the two-step binding model and our data can also be seen for the A3G mutants (Supplementary Fig. 4).

Using the same experimental procedures and analysis methods, we also find these kinetic rates for the A3G mutants (Fig. 4c–e). Both CTD mutants (IY and RDDQ) exhibit behavior very similar to that of WT, suggesting that CTD-mediated oligomerization does not significantly alter the binding kinetics of A3G. The NTD mutant (FW), however, binds to ssDNA at approximately three times the WT rate and oligomerizes nearly three times more slowly. Furthermore, although the total amount of A3G that is able to dissociate ($A_{dis}$) varies with incubation time and is greatly increased for the FW mutant, the rate at which it dissociates from

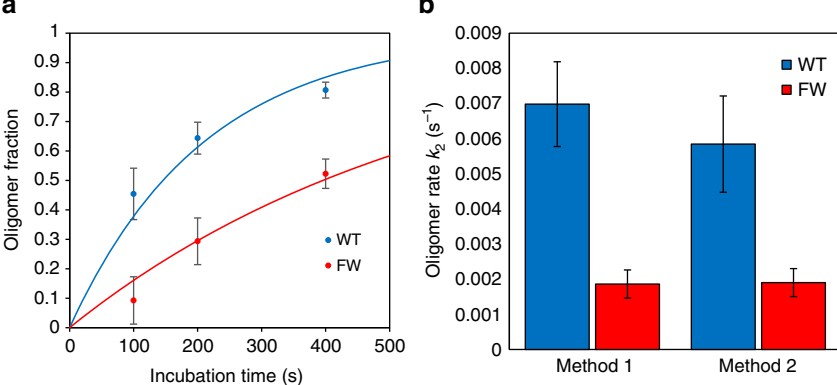

**Fig. 5** Measured oligomerization rate based on A3G in a dissociable state over time. **a** Fraction of bound A3G in the stable oligomer state as a function of incubation time at 50 nM. Values are calculated by dividing the amount of A3G that unbinds in buffer by the original amount of A3G after incubation ($A_{dis}$ and $A_{total}$ from the fitting of dissociation curves). WT A3G transitions to the oligomer state more quickly than FW mutant. An exponential decay function is fit to the data points to find an oligomerization rate. **b** Comparison of calculated oligomerization rate $k_2$ using the slow rate from the binding curve fit (data from Fig. 4) and this second method examining the amount of A3G that unbinds as a function of time. All *error bars* are SEM based on at least three independent experiments per incubation time

ssDNA is constant and equal for both WT and mutant A3G. In comparison with previous measurements of WT A3G binding[47], we obtain qualitatively similar results. However, due to improvements in protein purification and handling we observe greatly increased binding activity consistent with an increased active protein concentration. In addition, the technical advances to our instrumentation discussed above allow for direct observation of binding activity, which may avoid the introduction of artifacts introduced by long delays between observations and complications due to dsDNA reannealing.

**Alternative method confirms A3G oligomerization kinetics**. As compared with WT A3G, a large fraction of FW mutant A3G is still able to dissociate from the ssDNA after incubation (Fig. 3b). For example, after a 200 s incubation, less than half of the bound A3G is able to dissociate for both WT A3G and the CTD mutants, whereas more than two-thirds of the bound A3G dissociates for FW. We measured the total fraction of A3G that remains stably bound as function of incubation time ($A_{total}$ minus $A_{dis}$, divided by $A_{total}$) for both WT and FW mutant A3G (Fig. 5a). We fit a single exponential rate to the stabilization of bound protein over time (*red line*). This method returns similar results to the exponential fits described above, with the FW mutation reducing oligomerization rate by a factor of three (Fig. 5b). This independent measurement supports our model that the secondary binding process observed in the A3G binding curves is in fact the stabilization of A3G binding through oligomerization. In addition, from these curves it is clear that most if not all A3G is transiently bound when it initially binds at time zero. This supports a binding system in which the majority of free protein is in a monomeric state, as was previously observed for both WT and FW mutant A3G using AFM imaging[30], and that the formation of dimers and larger oligomers occurs primarily while bound to ssDNA.

**A3G dimerization prevents saturating binding of ssDNA**. The total fractional A3G binding at long times exhibited by the FW mutant, which inhibits N-terminal dimerization, also differs from that observed for WT and CTD mutant A3G. For 50 nM FW, nearly all of the ssDNA is bound by A3G during the fast initial binding, with only a small amount of additional association at longer times due to oligomerization (Fig. 6a). Based solely on the values of $k_1$ and $k_{-1}$ determined above, we would expect A3G

binding and dissociation to first reach equilibrium (before the dimerization of any protein) when ~90% of the available ssDNA is bound. However, this large amount of fast binding is only observed for the FW mutant. In contrast, the fast initial binding of both WT and the CTD mutants only binds a fraction of the ssDNA. The oligomerization process, however, is much more pronounced and results in the additional binding of ~40% of the ssDNA. Even at long incubation times, both WT and CTD mutants fail to fully saturate the ssDNA (Fig. 6a). During FW binding, the total extension of the ssDNA is consistently reduced by 0.051 nm bp$^{-1}$, with a standard deviation between experiments of just 0.003 nm bp$^{-1}$. For WT in contrast, the total extension change at long incubation times is both reduced to 0.037 nm bp$^{-1}$ and highly variable, with a SD of 0.011 nm bp$^{-1}$. The CTD mutants also display reduced and variable saturated binding similar to WT. Thus, some highly stochastic process that is dependent on NTD-mediated dimerization must be suppressing further binding of A3G.

To confirm the observed difference in the ability of WT and FW A3G to saturate ssDNA, we also performed a control experiment in which we observed ssDNA binding with equal concentrations of both WT and FW mutant A3G (both 50 nM) present in the sample (Fig. 6b). As expected, we see evidence that the two A3G variants both initially bind the ssDNA with an increased rate of binding higher than that exhibited by 50 nM of either variant alone. However, the total binding stops short of the saturating limit displayed by FW binding and even reduces at long times towards levels typical of WT binding alone. This trend indicates that WT A3G is able to displace the FW mutant at long times due to its increased rate of oligomerization as WT dimers remain bond for much longer than WT and FW monomers. The most likely mechanism to prevent saturated binding, even at high concentration of free protein, is the occlusion of binding sites. Shortly after A3G binds ssDNA, dimers start to form randomly along the length of the ssDNA strand. The locations may be inefficiently spaced, preventing the occupation of all possible binding sites. In addition, if A3G oligomers have reduced mobility along the ssDNA due to their increased binding affinity, the redistribution of protein along the ssDNA may be sufficiently slow to prevent saturated binding. In contrast, as the FW mutant remains in a monomeric form for significantly longer than WT A3G, the monomers can easily accommodate additional protein binding, allowing for saturated binding on short timescales (Fig. 6c). The same mechanism most likely accounts

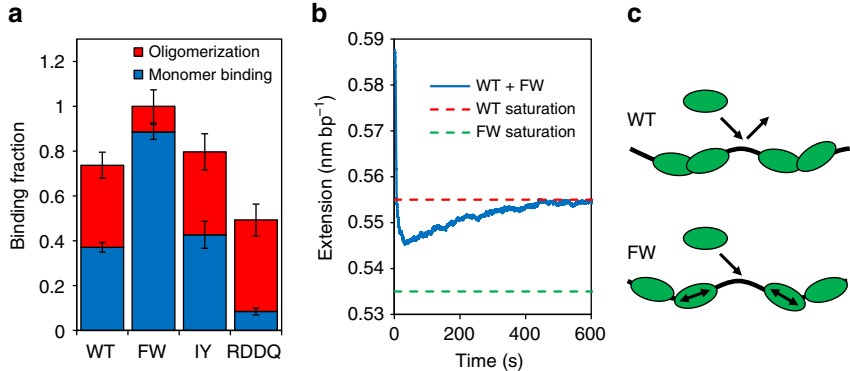

**Fig. 6** Effects of A3G oligomerization on saturated binding. **a** Average quantities of A3G binding due to fast initial binding (*blue*) and slow oligomerization (*red*) for WT A3G and the three mutants. Values are based on $A_{fast}$ and $A_{slow}$ obtained from fitting two state binding model. Binding amplitudes are normalized based on total binding by FW mutant at saturation. *Error bars* are SEM for at least three measurements. **b** Measured extension of ssDNA due to simultaneous binding of both 50 nM WT and FW A3G. The initial binding occurs at a faster rate than that due to either 50 nM WT or FW alone and results in an extension change in between that due to either WT or FW binding alone. Over time, the extension change decreases as more stably bound WT A3G replaces the more transient FW A3G. **c** Schematic showing how the formation of A3G oligomers with reduced mobility can prevent full saturated binding of ssDNA substrate. This effect is observed for both WT A3G and the CTD mutants. In contrast, for the NTD (FW) mutant, bound A3G remains in a monomer state for a significantly longer period of time and is able to saturate all binding sites through restricted diffusion before forming oligomers. This model is consistent with previous measurements showing altered sliding and jumping of FW mutant A3G as compared to WT[22]

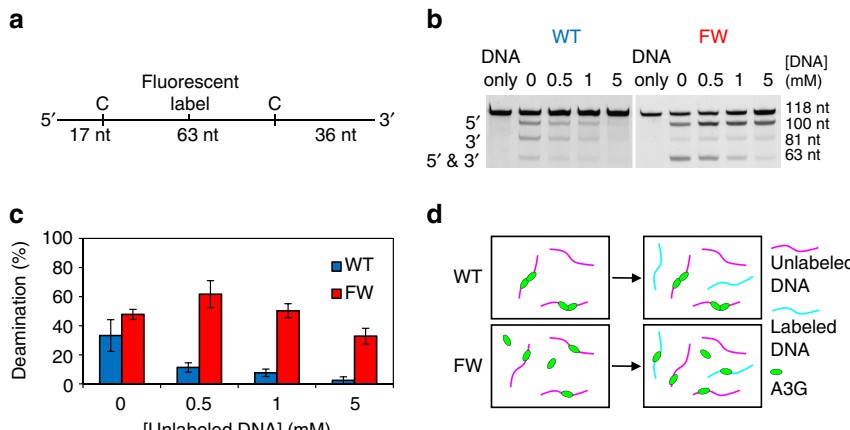

**Fig. 7** Comparison of deaminase activity of WT and FW mutant A3G. **a** Labeled ssDNA construct 118 nts in length with potential deamination sites 17 nts from the 5′-end and 36 nts from the 3′-end. The 63-nt segment between the deamination sites contains a fluorescently labeled thymine. After deamination, the label will be on a 100-, 81- or 63-nt-long strand if the 5′-site, 3′-site or both sites are cleaved, respectively. **b** Gel electrophoresis used to determine deaminase activity of WT (*left*) and FW mutant (*right*) A3G. A3G (25 nM) is preincubated with a 69-nt unlabeled ssDNA construct without deamination sites for 3 min before the labeled ssDNA is added at 100 nM and allowed to deaminate for 10 min. In lanes 2 through 5, a concentration of 0, 0.5, 1 and 5 mM unlabeled DNA is used. Lane 1 is a control with the labeled ssDNA construct only. Labeled ssDNA with the 5′-site, 3′-site or both sites deaminated appear in distinct bands. **c** Percentage of potential sites deaminated as a function of unlabeled DNA concentration for both WT and FW mutant A3G. Error bars show SD of at least three independent experiments. Preincubating WT A3G with unlabeled DNA before the introduction of the labeled DNA significantly reduces deaminase activity. In contrast, FW mutant A3G continues to deaminate the labeled DNA even when preincubated with large amounts of unlabeled DNA. Results indicate that the deaminase activity of A3G is strongly repressed by A3G oligomerization. **d** Representation of effect of A3G oligomerization on catalytic activity. During pre-incubation, WT A3G forms dimers, which inhibits the deamination of the subsequently added labeled DNA. In contrast, the FW mutant A3G remains in the monomeric state during pre-incubation and retains its deaminase activity

for the three-fold higher fast on rate $k_1$ for A3G FW, compared with the WT and CTD mutant A3G proteins. The appearance of immobile A3G dimers on ssDNA effectively slows down the association of incoming monomers, which would have to perform several binding attempts before a successful association event. This in turn effectively decreases the apparent $k_1$ of the dimerizing A3G WT proteins relative to the non-dimerizing FW proteins. It is important to note that in the virion the limited number of A3G monomers packaged would prevent such saturated binding from occurring. However, these results further demonstrate that A3G dimers bind ssDNA in a distinct manner

from A3G monomers both in terms of binding affinity and mobility.

**A3G deamination of ssDNA is inhibited by oligomerization.** The observed slow A3G oligomer dissociation from ssDNA, as well as the reduced oligomer sliding mobility suggested by our results would greatly impact A3G function. Indeed, the ability of A3G molecules to unbind and rebind an ssDNA substrate multiple times, as well as to slide along ssDNA in its bound state, is expected to be critical for just a few A3G molecules to be able to

deaminate about ~1000 sites on the ~9-kb-long HIV-1 genome, whereas the single stranded base pairs of the (−)DNA are exposed during RT. Thus, we would expect, based on the single molecule measurements presented above, that the oligomerization of A3G would inhibit its ability to efficiently search for deamination motifs within the viral genome. We developed a biochemistry assay to test this hypothesis. A 118-nt long labeled ssDNA construct was created with two potential deamination sites as previously described[22] (Fig. 7a). The two deamination sites were used to account for the different specific activities of WT and FW mutant A3G across the length of an ssDNA molecule[30]. The labeled ssDNA was incubated with either WT or FW mutant A3G for 10 min. After stopping the reaction, the DNA was treated with Uracil DNA glycosylase and NaOH to cleave the DNA at deamination sites and then DNA fragments were resolved by denaturing gel electrophoresis. Under these conditions, WT and FW mutant A3G exhibit comparable deamination activity (Fig. 7b). This result confirms that oligomerization is not required for deamination, as the FW mutant's inhibited dimerization does not diminish its deamination ability. To induce oligomerization before deamination, we also pre-incubated A3G with unlabeled ssDNA without the A3G 5'CCC deamination motif. This unlabeled DNA acted as a trap for the enzyme as we added it in increasing amounts to the enzyme and incubated the mixture for 3 min before the introduction of the labeled DNA. This timescale was chosen in accordance with the rate of oligomerization ($k_2 \approx (150\,\text{s})^{-1}$) observed in the single-molecule experiments, such that WT A3G will be able to form dimers but A3G FW ($k_2 \approx (500\,\text{s})^{-1}$) will not.

We find that the catalytic activity of WT A3G on the labeled DNA is reduced by threefold after pre-incubation with the trap DNA at low unlabeled DNA concentration (0.5 mM) and by 13-fold at high unlabeled DNA concentrations (5 mM), almost entirely suppressing A3G deamination activity (Fig. 7c). In contrast, FW mutant A3G, which remains primarily monomeric during the incubation period, retains its catalytic activity even when pre-incubated with high concentrations of unlabeled DNA (Fig. 7c). These results indicate that the oligomerization of A3G (specifically NTD-mediated dimerization) adversely affects its efficiency as a deoxycytidine deaminase. In particular, once WT A3G oligomerizes on non-specific ssDNA, it is unable to cycle between substrates and deaminate specific ssDNA, consistent with our single molecule binding results. In contrast, if A3G remains in a monomeric state (as FW mutant does), A3G can quickly cycle between ssDNA substrates in order to find and deaminate the labeled ssDNA (Fig. 7d). In addition, these results provide further indirect evidence that oligomerization suppresses A3G's mobility along ssDNA. Freely sliding A3G should be able to dissociate from these short ssDNA fragments by quickly reaching the end of the construct. Furthermore, we would expect some A3G dimers to transfer ssDNA strands via collisions and competitive binding between the many ssDNA molecules and then deaminate the labeled DNA if sliding mobility were fully conserved. Instead, the nearly full suppression of A3G deaminase activity suggests both cycling between substrates and sliding mobility are in fact partially reduced for WT A3G once dimers are formed. The results also help explain the multiple rates of deaminase activity previously observed for A3G[30, 50] as catalytic activity is decreased at longer times due to increasing oligomer formation.

## Discussion

In our single-molecule experiments, we make five major observations as follows: (1) Both CTD mutations studied (IY and RDDQ) exhibit similar binding kinetics as WT A3G, despite being deficient in forming large oligomers as observed under AFM. (2) After incubation, some fraction of bound A3G dissociates from ssDNA at a single, constant rate ($k_{-1} \approx (70\,\text{s})^{-1}$) for all A3G variants (Fig. 4d). The remaining bound protein does not dissociate during our observation time (up to 1000 s). (3) Mutating the NTD decreases the rate of formation of stable oligomers by a factor of 3 as measured independently by fitting the two-state binding model to the binding curves (Fig. 4f) and measuring the fraction of bound protein in the non-dissociative state as a function of incubation time (Fig. 5). (4) The binding of WT and CTD mutant A3G to ssDNA does not reach full saturation as compared with the binding of NTD mutant A3G (Fig. 6). (5) The NTD mutant A3G initially binds ssDNA at three times the rate as WT A3G at the same protein concentration (Fig. 4c). Combining these results, we can build an A3G binding model that clearly differentiates the role of both the NTD and CTD in the formation of stable oligomers and in the conversion of active A3G enzymes into slow ssDNA-binding proteins.

As disrupting the CTD dimerization interface does not significantly alter the observed binding kinetics of A3G, we conclude that the stabilization of A3G on ssDNA is primarily dependent on NTD-mediated dimerization. Thus, any further oligomerization (such as two NTD based dimers forming a tetramer through the CTD) would not be observed by our instrument, which directly measures binding kinetics on the timescales of 100 ms to 1000 s, as dimers already do not dissociate on this timescale. Thus, although WT A3G can form larger oligomers than the CTD mutants (as seen in our AFM results), the presence of one fast and one slow component of binding on this timescale is not significantly altered as long as NTD interactions and the A3G's ability to dimerize are preserved.

We expect the dissociation rate of an A3G molecule to depend on its degree of oligomerization (e.g., tetramers bind longer than dimers, which bind longer than monomers) as larger oligomers have additional binding sites that interact with ssDNA. The single dissociation rate we observe at $k_{-1} = 0.013\,\text{s}^{-1}$ must be the rapid dissociation of A3G monomers, as this rate is present for all A3G variants, and no faster dissociation is observed that could be attributed to monomers. The dissociation of dimers and larger oligomers must occur at a longer timescale not observed during the timescale of our experiments (up to 1000 s). Thus, although monomeric A3G remains bound to ssDNA on the timescale of <100 s, dimerization reduces the rate of dissociation by at least an order of magnitude. This scheme is further supported by the effect of NTD and CTD mutation on A3G oligomerization. If WT A3G initially bound ssDNA as an NTD-mediated dimer, further oligomerization (tetramers, etc.) would occur due to CTD–CTD interactions. Neither CTD mutation, however, is able to significantly disrupt the observed dimerization. Instead, our results are again consistent with A3G molecules binding initially as monomers and then forming NTD-mediated dimers at a rate of $k_2 = 0.007\,\text{s}^{-1}$. In contrast, when the NTD dimerization interface is disrupted, dimers form at a reduced rate $k_2 = 0.002\,\text{s}^{-1}$ due to CTD interactions (or possibly the mutated NTD retains some dimerization ability and CTD interactions are even slower). Thus, for WT A3G, dimerization alone, primarily facilitated by the NTD, is a sufficient degree of A3G oligomerization on ssDNA for this protein to become very slow and immobile.

Dimerization also affects the ability of A3G to saturate all possible ssDNA binding sites. The high mobility of the A3G monomer allows for rapid diffusion along the ssDNA strand, which is necessary for a small number of A3G subunits to deaminate the viral (−)DNA before the synthesis of the (+) DNA. Conversely, the slow sliding and dissociation of A3G dimers would greatly diminish this scanning ability. Thus, the NTD mutant (FW) is able to more fully saturate ssDNA during

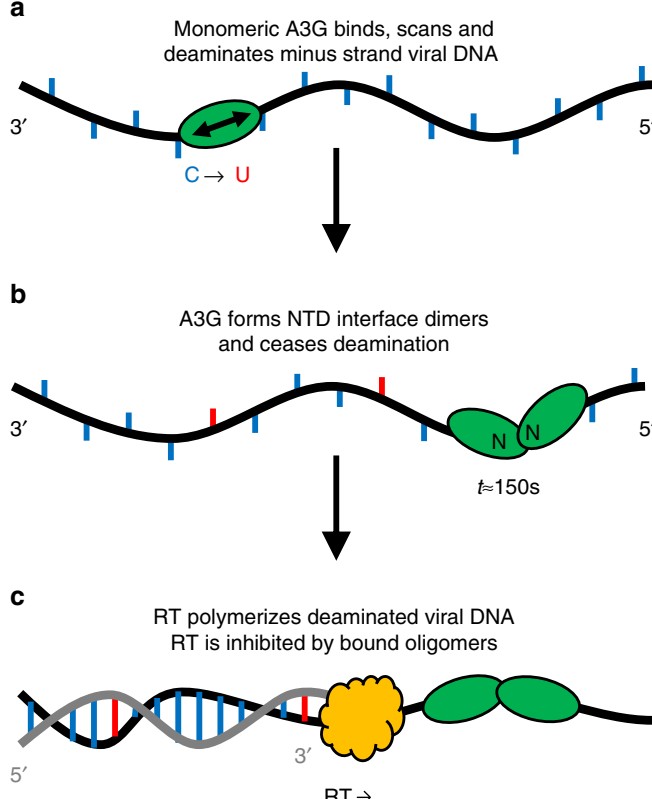

**a** Monomeric A3G binds, scans and deaminates minus strand viral DNA

$C \rightarrow U$

**b** A3G forms NTD interface dimers and ceases deamination

$t \approx 150 s$

**c** RT polymerizes deaminated viral DNA
RT is inhibited by bound oligomers

RT →

**Fig. 8** Schematic of two mode inhibition of HIV-1 replication by A3G. **a** Monomers bind and slide along the minus strand viral DNA, allowing for rapid deamination. **b** After ~150 s, bound A3G monomers form dimers through NTD interactions. Dimers do not slide along ssDNA, inhibiting catalytic activity. **c** RT polymerizes the complementary plus strand viral DNA, including G to A substitutions at sites of deamination. The formation of the full plus strand is also inhibited by bound A3G oligomers

initial A3G binding as the bound A3G are able to rapidly accommodate additional subunits. In contrast, WT A3G begins to form slow dimers much sooner, inhibiting saturated binding. As a result, both WT and CTD mutant A3G saturate a significantly smaller fraction of the ssDNA binding sites as compared with the NTD mutant. Finally, the inhibition of further A3G binding due to the formation of slowly diffusing oligomers could also explain the threefold increase in the rate of binding exhibited by the NTD mutant. In the presence of dimers and larger oligomers bound to ssDNA, monomers in solution are unable to access occluded binding sites, reducing the effective binding rate. It is important to note that, in the virion, only a small number of A3G subunits are present and such interactions are greatly reduced.

Our results compare well with previous studies on both A3G binding and deaminase activity when the important differences between single molecule and bulk assays are taken into account. In particular, bulk kinetics studies on ssDNA binding and dissociation, in which many ssDNA constructs compete for A3G binding, may show different kinetics than our method, which more closely resembles *in virio* conditions with a single viral DNA template. A previous study that examined both WT and FW mutant A3G found that the FW mutant bound ~50% more ssDNA than WT A3G at a saturating concentration of 1 μm[30]. In addition, a large fraction of the bound WT A3G was found to be unable to dissociate once bound, in agreement with our data. Although there have been different views on whether A3G's primary active enzymatic form is monomeric or

oligomeric, multiple single molecule (Fig. 4) and AFM (Fig. 1) measurements, as well as a bulk deamination assay (Fig. 7) presented here, conclusively show that the monomeric form of A3G is catalytically active. This agrees with previous results showing that oligomerization-deficient mutants are still capable of deamination[30, 31, 42]. In addition, the deaminase activity of A3G has been shown to exhibit significant asymmetry both in binding conformation and directional catalytic specificity on ssDNA substrates. It has been shown that A3G more readily deaminates when sliding towards the ssDNA substrate's 5′-end with the CTD leading[30, 51]. However, A3G's intrinsic asymmetry (with a distinct NTD and CTD) is lost when forming an NTD to NTD dimer. In addition, it has been shown that both WT and FW mutant A3G fail to deaminate a small region close to the 3′-end[30], in agreement with the predicted favorable binding state with the CTD bound facing towards the 5′-end[51, 52]. This scheme also requires monomeric deaminase activity from WT A3G. Finally, with new evidence that A3G oligomerization affects its binding to ssDNA and its catalytic activity, further research can focus on identifying the exact nature of the A3G deaminase-independent mode of HIV restriction. It has been shown that high concentrations of A3G can in fact inhibit RT[26]. Using the techniques and A3G mutations discussed in this work, the oligomeric state of A3G can be tightly controlled, in order to determine its role in altering RT.

The observation of dual timescales for fast initial binding of A3G monomers and slow dimerization helps resolve the major apparent conflict of the roadblock model for A3G's deamination-independent inhibition of HIV-1 replication. Thus, interactions between the catalytically inactive NTDs regulate the dual functions of A3G (Fig. 8). Although only $7 \pm 4$ A3G monomers are incorporated into a single HIV-1 virion in the absence Vif[53], the effective A3G concentration is of the order of 10 μM due to the virion's small volume[20], resulting in near instantaneous binding to minus strand viral DNA. While bound for on average 70 s $(k_{-1})^{-1}$, monomers rapidly slide in a localized area to search for a deamination motif and repeatedly dissociate and reassociate along the single-stranded $(-)$DNA to restart the search process and enable deamination of multiple deoxycytidines. After several minutes ($k_2 \approx (150 \text{ s})^{-1}$ at saturating quantities of A3G), the A3G dimerizes and remains stably bound. Thus, A3G monomers are able to bind, scan the $(-)$DNA for deamination sites, and then dissociate or jump to a new regions of ssDNA for several cycles before forming dimers, ensuring efficient deamination of the entire viral genome. Although the dimer's extremely slow dissociation and reduced sliding mobility inhibits A3G's enzymatic function, these properties also allow A3G dimers to act as a roadblock for RT. As only dimers, rather than large oligomers, are required to create a roadblock, effective roadblocks are much more likely to form given the limited number of A3G molecules available. Multiple roadblocks can even be formed in order to further delay RT and the formation of $(+)$DNA, in order to allow the remaining A3G monomers to more fully scan the entire $(-)$DNA substrate. Thus, NTD-mediated dimerization enables A3G to act as a fast deaminase at short timescales and a RT inhibitor at long timescales, even with the low number of A3G molecules found in the virus.

## Methods

**A3G purification and preparation.** A3G protein was expressed in baculovirus and constructed by PCR amplification of the coding portion of A3G from IMAGE clone 4877863 (ATCC)[30]. Recombinant baculovirus production for expression of glutathione *S*-transferase (GST)-A3G in Sf9 cells was carried out using the transfer vector pAcG2T (BD Biosciences). Sf9 cells were infected with recombinant virus at

a multiplicity of infectionof 1 for 72 h. A3G was purified in the presence of RNase A and the GST tag cleaved on the affinity column. The A3G is ~95% pure by SDS-polyacrylamide gel electrophoresis.

**AFM imaging**. Purified A3G was diluted to a concentration of 70 nM in a buffer of 10 mM Tris, 2.5 mM NaCl and incubated with linearized m13 ssDNA. After allowing A3G binding to ssDNA for 10 min, the sample was deposited on an APTES coated mica surface and imaged in liquid using an AFM with peak force tapping mode[54, 55]. For each individual ssDNA strand, a height threshold was applied in order to differentiate bound protein from bare ssDNA. A Gwyddion segmentation function was used to group neighboring pixels above the threshold into individual protein clusters. The integrated volume of each cluster, based on height and pixel area, was used to determine the total number of monomers it contained assuming protein cluster size is proportional to number of bound subunits.

**Single-molecule experiments**. Experiments were performed using biotinylated bacteriophage λ-DNA (48,500 bp) attached at each end to streptavidin coated polystyrene beads in a buffer containing 10 mM HEPES and 50 mM Na$^+$ at pH 7.5 at 21 °C[56]. One bead was held by a micropipette tip which is moved using a piezoelectric stage while the other bead is held in a duel beam optical trap. Using the position of the micropipette tip and the deflection of the optical trap beams, both the force exerted on the DNA and its resulting extension are constantly measured. Purified A3G was dilated in buffer to concentrations ranging from 10 to 200 nM and flowed into the sample chamber. During A3G binding and dissociation, active force feedback, in which the position of the pipette tip was moved based on the force exerted by the optical trap, was used to maintain a constant force of 80 pN. Each reported measurement was done on a separate DNA molecule to obtain independent measurements of the quantities of interest.

**Deamination assay**. Deamination activity was measured using a labeled ssDNA construct 118 nt in length with potential deamination sites 17 nt from the 5′-end and 36 nt from the 3′-end (see Supplementary Fig. 5 for full sequence). Multiple deamination sites were used to remove any bias in the activity calculation should the enzyme prefer to deaminate one site more than the other. The 63 nt segment between the deamination sites contains a fluorescently labeled thymine. 25 nM A3G was preincubated with unlabeled ssDNA 69 nt in length in concentrations up to 5 μM for 3 min in a buffer containing 10 mM HEPES, 50 mM Na$^+$ and 1 mM dithiothreitol at pH 7.5 at room temperature (21 °C). The labeled DNA was then added at a concentration of 100 nM and incubated for 10 min at 37 °C. Reactions were stopped by phenol:chloroform extraction and formation of uracils was detected by incubation of ssDNA with Uracil DNA N-glycosylase (New England Biolabs) and heating in the presence of NaOH. This treatment fragmented the ssDNA at deamination sites. Gel electrophoresis of the sample produced four bands corresponding to labeled segments 118, 100, 81 and 63 nt in length indicating no deamination, 5′-deamination, 3′-deamination, and 3′- and 5′-deamination, respectively.

**Data availability**. The data supporting the findings of this study are available within the paper and its Supplementary Information files and available from the corresponding author upon reasonable request.

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

## Acknowledgements

This work was supported by National Institute of Health grant GM072462 (MCW), National Science Foundation grant MCB-1243883 (MCW) and Canadian Institutes of Health Research grant MOP137090 (L.C.). We thank Micah McCauley for preparation of biotinylated bacteriophage λ-DNA and Judith Levin for valuable discussions.

## Author contributions

M.M., M.C.W., L.C. and I.R. designed the experiments. M.M. performed single-molecule experiments and analyzed the data. R.H. performed AFM experiments. L.C. performed deamination experiments. Y.F. and L.C. prepared the proteins. M.M., M.C.W, L.C. and I.R. wrote the manuscript.

## Additional information

**Competing interests:** The authors declare no competing financial interests.

