## [Peer Review File · Nature Communications]

Reviewers' comments:

Reviewer #1 (Remarks to the Author):

Williams and coworkers perform single-molecule 'optical tweezers' studies on the human enzyme APOBEC3G, which is a potent retroviral restriction factor. This enzyme contains tandem cytidine deaminase domains (CDA1 and CDA2), which have separate functions in nucleic acid binding, catalysis and interaction with viral antagonists such as HIV-1 Vif. Previously, Williams and colleagues measured the kinetics of DNA binding by A3G using optical tweezers on stretched dsDNA at a constant force beyond 60pN where the DNA is single-stranded. This study revealed 1- A3G binds ssDNA with biphasic association and dissociation kinetics; 2- the microscopic rate constants were determined for the initial binding as well as a unimolecular conversion into a slowly dissociating state, which was postulated to form a steric 'roadblock' to viral reverse transcriptase (RT) (Chaurasiya et al, Nature Chemistry, 2013). 3- residues F126W127 in CDA1, are responsible for the slow binding mode. This is consistent with prior bulk biochemistry (SEC-MALLS) and atomic force microscopy (AFM) studies carried out on full-length enzyme and the FW variant. Moreover, a recent crystal structure of the CDA1 domain of rhesus macaque A3G indicates F126W127 of CDA1 lines the dimer interface.

The present manuscript is a follow-up study, looking at the role of surface residues of CDA2 of A3G: residues I314Y315 and R313D316D317Q318) that have been implicated in oligomerization and therefore, may mediate the slow oligomerization of A3G. The authors find both sets of CDA2 mutations –the IY double or RDDQ tetra mutants -- exhibit similar binding kinetics to wild-type A3G. They conclude that stabilization of A3G on DNA (the slow binding mode) is primarily mediated by CDA1, consistent with what was previously reported in the aforementioned manuscript by Chaurasiya et al.

In addition, the authors show the slow -phase binding limits the ability of A3G to saturate all possible sites. In this way, CDA1 poses a paradox: it is required for dimerization and packaging of A3G into virions but at the same time limits the amount of A3G that can be bound to 1st strand cDNA, the target of deamination. They posit a model, based on the kinetic constants extracted from the present manuscript and prior publications, that A3G binds as a monomer, undergoes rapid restricted diffusion to hypermutate viral genomes and then slowly converts into RT roadblocks at a late step after most sites are edited.

The use of optical tweezers to measure A3G binding kinetics and the observation that the FW mutation in CDA1 are responsible for the conversion into the slow binding form was reported by the Williams lab several years ago (Chaurasiya et al, 2013). At that time, using optical tweezers to study A3G was entirely novel, provided new insights into the timescales of oligomerization changes induced by DNA binding and mapped requirements for the slow phase to CDA1 of A3G. The present manuscript is more incremental and more suitable for publication in a specialized journal.

It could be improved if the following questions are taken into consideration:

1-page 2, 2nd paragraph of the introduction, the authors state three-dimensional search mechanism termed 'sliding, jumping and intersegmental transfer' is employed by A3G. These processes are not a three-dimensional diffusive process but rather 'restricted diffusion in 1 or 2 dimensions' (as elaborated by von Hippel and colleagues.)

2-The bimolecular association rates for wild-type A3G (k_1) appear to be a log-unit faster in the present manuscript compared to what was previously reported in the aforementioned publication by the same lab Chaurasiya et al ($0.001 \text{ s}^{-1} \text{ nM}^{-1}$ vs $0.00015 \text{ s}^{-1} \text{ nM}^{-1}$). Why the discrepancy?

3- FW mutant A3G more readily saturates ssDNA than wildtype A3G and these differences are

attributed to differences in the fast binding step. The k_1 for FW mutant A3G is 3-fold greater than that for wild-type. Measuring this from the fitted slope of Fig 4a seems to provide a reliable estimate; however, I'm not convinced the measure of k^{-1} from the y-intercept is reliable given scatter in fast step. It would be useful if plots of k_{fast} vs A3G concentration for mutants were provided, in addition to wildtype.

4-Though one could in principle obtain k_1 and k^{-1} from the concentration dependence of k_{slow} , these data are also problematic. At the concentration range used, there is not much variation in k_{slow} (Fig 4b) which might make the extraction of k^{-1} unreliable. In contrast, the asymptote of Fig 4b is the sum of k_2 and k^{-2} which is well determined.

5-Legend, Figure 6: with respect to the FW variant of A3G, the authors state 'the mobile monomers can slide to accommodate additional binding, resulting in additional binding'. I disagree. The authors do not directly measure sliding and, moreover, the conversion to the slow phase (k_2) is actually 3-fold slower for the FW mutant. I think the authors correctly explain the phenomenon in the discussion: the sum of k_1, k^{-1} determine the approach to equilibrium and this is faster for the FW mutation. It is reasonable to interpret the reduced rate of conversion into the 'slow' form of A3G for this mutant as allowing for rapid restricted diffusion or jumping/sliding as a mechanism to cover all potential A3G binding sites on DNA.

Reviewer #2 (Remarks to the Author):

In their manuscript, Morse and coworkers describe intriguing new data and analysis on the protein APOBEGEC3G and several mutants, the binding of which to ssDNA they study with optical tweezers and atomic force microscopy. They find further evidence that the binding of the protein is a two-step process: first the protein binds as a monomer to the DNA and subsequently, on a much slower time scale, can form dimers, that bind much more tightly. These findings provide crucial new insights in the fascinating ant-HIV activity of this human protein: it inactivates the HIV genome in two ways: by chemically modifying it (for which it needs to be highly mobile) and by forming roadblocks for reverse transcriptase (for which it needs to be tightly bound). Overall this is an exciting manuscript, with sound data and analysis, which could very well fit into Nature Communications. Before I can unequivocally make that call, I have a couple of suggestions to improve the manuscript as well as a couple of questions on interpretation and modeling.

More important aspects

1) The manuscript builds very strongly upon an earlier study by the same group (Nature Chem 2014, ref 43). In principle nothing wrong with that. But I think that the current data is not optimally described in connection with the previous data. E.g. Just above eq. 2: why can the time dependence of f now be measured directly and not before? What is that a technical innovation? Furthermore, the new data for WT should be compared to the similar data in 43. A bit more discussion (in intro and discussion) about what is new would help the reader (and there is clearly enough novelty). Also it should be more clearly stated that equations 2-5 are basically the same used in ref. 43.

2) I keep on breaking my head on the model used by the authors. Key thing I would like to understand (and explained / discusses) is the fact that the second step, the dimerization, only seems to depend upon the amount (ratio) of A3G bound to the DNA. I would intuitively expect it to depend upon the square of the amount (ratio), since two monomers first have to meet (by diffusion along the DNA) before they can dimerize. Or does this have to do with the time scales of the processes. Please explain and discuss!

3) I think it would be nice if the authors show all data and exponential fits (in supplemental info). Only four examples are shown in figure 3 (only 3 fitted...), while there are 4 concentration * 4

mutants * 2 (dissociation + incubation) = 32 curves. Idem for the likes of figure 4ab for the other mutants: it would be great to see these curves.

4) Basically the authors perform their experiments at a pretty high force: 80 pN. Can they exclude that force affects binding & release of A3G and its mutants? One would expect so given the shortening of ssDNA upon A3G binding. This would mean that there is a mechanical component to A3G binding to ssDNA, which is counteracted by applying 70 pN! The explanation given in p. 4 that A3G wraps DNA is in this respect a bit scary: DNA wrappers like E. coli or mitochondrial SSB do not bind to ssDNA under substantial tension.

Minor aspects:

I found the the different abbreviations for the mutants somewhat confusing: on the one hand NTD/CTD , on the other FW, WT, IY and RDDQ.

The authors call their tweezers experiments several times single-molecule experiments. I find this a bit confusing: sure, 1 dsDNA molecule is used, but the binding of many (in fact: how many??? thousands, ten thousands) molecules of A3G is measured, not single binding events.

p.1 "fast and slow binding state". I found this confusingly colloquial, particular here in abstract. Fast and slow should be between quotes, or explained a bit more precise. What about mentioning it a 2-step binding process, initial reversible binding followed (on a slower time scale) by oligomerization resulting in very stable binding.

p. 6 "As compared to WT A3G, a large..." The (or at least this) reader would be helped with a reference to fig 3B here.

p. 7 I found the paragraph on top a bit confusing. I guess that the main cause is "The total fractional A3G binding". If I understand it correctly, what the authors mean is at long time or after equilibration. Maybe it would be good to explicitly say this.

p. 7 "Indeed, as the time-scale for fast monomer association" I found this sentence very complicated and difficult to understand. Please break up and explain with more words, in more detail.

p. 10 Top paragraph. I found the use of "rapid movement" and "slow dimers" and "slow oligomers" and "slow binding kinetics of dimers" rather confusing. Maybe the authors could be more explicit in referring to a "slow/fast binding equilibrium" and "slow/fast diffusion".

p. 11 One but last sentence of discussion: "()DNA substrate" I guess this should be (-).

fig. 3 please indicate A3G concentration.

Reviewer #3 (Remarks to the Author):

The authors present an interesting story of dual role of the human APOBEC3G enzyme in inhibiting HIV reverse transcription. The described mechanisms are rather novel and interesting and, hence, worth publishing.

One of their tools is AFM, where they show that the wildtype forms a sequence of oligomers on SS DNA while several mutants exhibit reduced ability to oligomerize.

Unfortunately, there are no images from such experiments and the information as to how the volumes of the bound proteins were measured is limited (Gwyddion segmentation function is not an adequate description of method). Such data and information should at least be part of supplemental materials. In addition, wouldn't it be possible to examine the binding of the enzymes

as a function of concentration and time under the AFM to corroborate the results of the tweezers? Events may be too fast for AFM imaging but they can probably still be captured with the help of substrate binding that may, in principle, slow dissociation significantly.

Response to reviewers

Reviewer #1

Previously, Williams and colleagues measured the kinetics of DNA binding by A3G using optical tweezers on stretched dsDNA at a constant force beyond 60pN where the DNA is single-stranded. This study revealed 1-A3G binds ssDNA with biphasic association and dissociation kinetics; 2- the microscopic rate constants were determined for the initial binding as well as a unimolecular conversion into a slowly dissociating state, which was postulated to form a steric 'roadblock' to viral reverse transcriptase (RT) (Chaurasiya et al, Nature Chemistry , 2013). 3- residues F126W127 in CDA1, are responsible for the slow binding mode [...] The present manuscript is a follow-up study, looking at the role of surface residues of CDA2 of A3G: residues I314Y315 and R313D316D317Q318) that have been implicated in oligomerization and therefore, may mediate the slow oligomerization of A3G. The authors find both sets of CDA2 mutations –the IY double or RDDQ tetra mutants -- exhibit similar binding kinetics to wild-type A3G. They conclude that stabilization of A3G on DNA (the slow binding mode) is primarily mediated by CDA1 , consistent with what was previously reported in the aforementioned manuscript by Chaurasiya et al.

Response: We would like to clarify a few points regarding the findings of this work in relation to those of Chaurasiya et al. It is correct that the biphasic relationship of A3G binding was first detailed in this previous work and that the measurements were made using an optical tweezers system. However, the measurements had significant technical limitations and left many issues unresolved concerning the mechanism through which A3G binding is stabilized. First, due to technical limitations, A3G binding could only be measured by allowing incubation for set periods of time and then slowly stretching and releasing the DNA (on the timescale of minutes) to observe dsDNA reannealing, limiting measurement accuracy. In contrast, we now observe A3G binding in real time with 10 ms time resolution at a single force without artifacts due to long pauses in data acquisition. This enables us to directly observe A3G binding, dissociation, and oligomer formation. In Chaurasiya et al, it was proposed that this second binding state was a result of A3G oligomerization, but the size of the oligomers was unknown. Although the A3G FW mutant was tested, the limited amount of protein available prevented characterization of its binding properties beyond the fact that it did not oligomerize. Here we fully characterize several A3G mutants, revealing for the first time that NTD dimerization alone is responsible for slow A3G binding as well as inhibition of deamination activity.

Reviewer Comment 1: page 2, 2nd paragraph of the introduction, the authors state three-dimensional search mechanism termed sliding, jumping and intersegmental transfer' is employed by A3G. These process are not a three-dimensional diffusive process but rather 'restricted diffusion in 1 or 2 dimensions' (as elaborated by von Hippel and colleagues.)

Response: Indeed von Hippel and colleagues were the first to describe facilitated diffusion. In the von Hippel and Berg 1989 J. Biol. Chem. paper they did not discuss 3-dimensional movements as we described in our introduction, but instead used the terms sliding, intra-domain dissociation and reassociation, and intersegmental transfer. In work that followed, von Hippel and Berg, Halford, and Marko published a series of papers examining how restriction enzymes located their target motifs. In this expanded and modified facilitated diffusion model, the authors described terms such as sliding, intersegmental transfer, and jumping and hopping instead of intra-domain dissociation and reassociation. They further go on to describe sliding as a one-dimensional search and hopping, jumping, and intersegmental transfer as a three-dimensional search. We have adopted this established terminology and clarified this in the introduction by including references to the von Hippel and Halford work.

Reviewer Comment 2: The bimolecular association rates for wild-type A3G (k_1) appear to be a log-unit faster in the present manuscript compared to what was previously reported in the aforementioned publication by the same lab Chaurasiya et al ($0.001 \text{ s}^{-1} \text{ nM}^{-1}$ vs $0.00015 \text{ s}^{-1} \text{ nM}^{-1}$). Why the discrepancy?

Response: Since the Chaurasiya et al paper we have modified our purification method and this has likely increased the active fraction of protein used in these studies. This is most readily apparent for the FW mutant, where in contrast to Chaurasiya et al. we now see comparable binding activity to WT. Additionally, as discussed above, the previous study had certain technological limitations. Since the binding of A3G could only be inferred from the ability of A3G to reanneal over the course of a several minute stretching curve, the dissociation of A3G over this timescale may have led to an under-measurement of bound A3G. Thus, while these new results compare favorably with the previous work qualitatively, the new quantitative measurements are derived more directly and should be considered more definitive results. A more direct comparison of the current results with the previous paper has been added to the manuscript at the top of p. 5 as well as the bottom of p. 6.

Reviewer Comment 3: FW mutant A3G more readily saturates ssDNA than wildtype A3G and these differences are attributed to differences in the fast binding step. The k_1 for FW mutant A3G is 3-fold greater than that for wild-type. Measuring this from the fitted slope of Fig 4a seems to provide a reliable estimate; however, I'm not convinced the measure of k_1 from the y-intercept is reliable given scatter in fast step. It would be useful if plots of k_{fast} vs A3G concentration for mutants were provided, in addition to wildtype.

Response: Our description in the manuscript of how exactly we obtain the fundamental rate constants was not clear and that paragraph has been completely rewritten on p. 6. The caption for figure 4 has also been updated. In short, the fits are not actually needed to find the elementary rate constants. This fitting was necessary in the previous work, but our ability to track binding in real time now enables us to directly observe all the rates in a single experiment.

For example, k_{-1} is measured directly from dissociation measurements at $c=0$. For other concentrations, we use multiple measurements of k_{fast} and k_{slow} to obtain the values shown in panels c-e. Once we obtained the rates, we then plotted the 'fit' lines in figures 4a and b to show agreement between the model and data. We have included a new figure S4 to show the concentration dependence of k_{fast} for the A3G mutants. However, the agreement between the dissociation rates is directly observed, not inferred from a fit (note the overlap of value at $c=0$ in Fig. S4). We also know this value quite precisely. While the error bars are too small to see in panel a, the same standard error is also shown in panel d. For comparison, we tried fitting the k_{fast} data directly with a linear fit rather than first deriving the values and then plotting. The results are nearly identical (see blue and black line in figure S4), providing further validation of our model describing the data.

Reviewer Comment 4: Though one could in principle obtain k_1 and k_{-1} from the concentration dependence of k_{slow} , these data are also problematic. At the concentration range used, there is not much variation in k_{slow} (Fig 4b) which might make the extraction of k_{-1} unreliable. In contrast, the asymptote of Fig 4b is the sum of k_2 and k_{-2} which is well determined.

Response: As stated above, k_{-1} is determined directly from A3G dissociation measurements, while k_1 is obtained directly from multiple concentration measurements of k_{fast} , so these values can be used directly and are not in any way fit to k_{slow} measurements. Thus, after using the previously determined k_{-1} and k_1 , we obtain k_2 directly from Eq. 4, where k_{-2} is negligible at all concentrations used, as described on p. 6. The upper limit of k_{-2} is determined in a separate experiment, as also described on p. 6.

Reviewer Comment 5: Legend, Figure 6: with respect to the FW variant of A3G, the authors state 'the mobile monomers can slide to accommodate additional binding, resulting in additional binding'. I disagree. The authors do not directly measure sliding and, moreover, the conversion to the slow phase (k_2) is actually 3-fold slower for the FW mutant. I think the authors correctly explain the phenomenon in the discussion: the sum of k_1, k_{-1} determine the approach to equilibrium and this is faster for the FW mutation. It is reasonable to interpret the reduced rate of conversion into the 'slow' form of A3G for this mutant as allowing for rapid restricted diffusion or jumping/sliding as a mechanism to cover all potential A3G binding sites on DNA.

Response: The FW mutant has been previously characterized by two of the authors of this manuscript (Feng and Chelico) in the publication Feng and Chelico, J. Biol. Chem., 2011. In this paper the authors found that the FW mutant did move on ssDNA by one-dimensional sliding, but was unable to undergo three-dimensional movements. WT A3G was able to move by one-dimensional sliding and three-dimensional jumping, but the reaction rate was slower than the FW mutant, consistent with the data presented in the manuscript and the above statement by the referee. We now reference the above work and use a statement similar to that presented by the reviewer in the legend to Fig. 6.

Reviewer #2

Reviewer Comment 1: The manuscript builds very strongly upon an earlier study by the same group (Nature Chem 2014, ref 43). In principle nothing wrong with that. But I think that the current data is not optimally described in connection with the previous data. E.g. Just above eq. 2: why can the time dependence of f now be measured directly and not before? What that a technical innovation? Furthermore, the new data for WT should be compared to the similar data in 43. A bit more discussion (in intro and discussion) about what is new would help the reader (and there is clearly enough novelty). Also it should be more clearly stated that equation 2-5 are basically the same used in ref. 43 .

Response: The previous work is now described in more detail and is directly compared with our new results on p. 5. The exact technical advancements are detailed in the description of the apparatus, in particular the 10 ms response time and 1 nm extension control. Also, while equations 2-5 are derived based on the same model (eq 1, which we do reference) they are slightly different than those used in the previous work as we can now observe binding directly rather than fitting to discrete data points based on set incubation times.

Reviewer Comment 2: I keep on breaking my head on the model used by the authors. Key thing I would like to understand (and explained / discusses) is the fact that the second step, the dimerization, only seems to depend upon the amount (ratio) of A3G bound to the DNA. I would intuitively expect it to depend upon the square of the amount (ratio), since two monomers first have to meet (by diffusion along the DNA) before they can dimerize. Or does this have to do with the time scales of the processes. Please explain and discuss!

Response: The dependence of oligomerization rate on the degree of DNA saturation with protein can depend on the protein concentration in many different ways, as was discussed, for example, in our work on the SSB T4 gp32 (Pant et al, JMB 2004). However, the two-state model presents the simplest approximation that fits the available data within our experimental accuracy. The conclusions we make based on the rates of the two-step processes, as well as their comparison between A3G mutants, should not depend on the details of a more complicated model, which we do not have sufficient data to describe. There is also difficulty in measuring oligomerization rates at low protein concentration as the fast initial binding rate decreases, the two rates are not easily separable (see 10 nM data in fig s2). This limitation prevents us from being able to determine a detailed model for the diffusional search used by A3G to form dimers. Fortunately, such data is not needed to determine the fundamental rates reported here, which describe for the first time the oligomerization states associated with the changes in A3G binding kinetics.

Reviewer Comment 3: I think it would be nice if the authors show all data and exponential fits (in supplemental info). Only four examples are shown in figure 3 (only 3 fitted...), while there

are 4 concentration * 4 mutants * 2 (dissociation + incubation) = 32 curves. Idem for the likes of figure 4ab for the other mutants: it would be great to see these curves.

Response: Each experimental condition is repeated multiple times, so the total set of binding curves is actually quite large. The examples shown in the paper are mainly for illustrating how the fundamental rate constants (k_1 , k_{-1} , k_2) are derived from experimental curves, which are our main results. However, we have added numerous more example curves in two new supplemental figures, including fits lines produced according to the two step binding model. These show more graphically the impact of concentration (fig S2) and mutation (fig S3) on A3G binding. As discussed in response to another reviewer above, we now also now show concentration dependence data for all mutants (fig s4).

Reviewer Comment 4: Basically the authors perform their experiments at a pretty high force: 80 pN. Can they exclude that force affects binding & release of A3G and its mutants? One would expect so given the shortening of ssDNA upon A3G binding. This would mean that there is a mechanical component to A3G binding to ssDNA, which is counteracted by applying 70 pN! The explanation given in p. 4 that A3G wraps DNA is in this respect a bit scary: DNA wrappers like E. coli or mitochondrial SSB do not bind to ssDNA under substantial tension.

Response: The total DNA extension change upon saturated A3G binding at 80 pN is quite small (at most 0.05 nm/bp), which is <10% of the ssDNA contour length at that force. This effect is much smaller than the highly ordered wrapping exhibited by E. coli SSB, in which 35 or 65 base pairs (the equivalent of 20-35 nm of ssDNA contour length) are wrapped around each SSB tetramer (Lohman et al. Annual Rev Biochemistry 1994. Raghunathan et al. NSMB 2007, etc). Therefore, despite the large ssDNA binding free energy of E. coli SSB, relatively low forces (~10pN) drive this protein off ssDNA. In contrast, the observed minor (but measurable) ssDNA shortening by A3G leads to a weak ssDNA binding force dependence, such that even at 80 pN its binding is almost as strong as without applied force. We now note this in the manuscript at the beginning of p. 5.

Minor aspects:

I found the the different abbreviations for the mutants somewhat confusing: on the one hand NTD/CTD , on the other FW, WT, IY and RDDQ.

Response: The FW mutant is an NTD mutant while both IY and RDDQ are CTD mutants. Ultimately we are more interested in what domain is responsible for oligomerization, though we still use the exact mutations in the figure legends. We have attempted to clarify both the mutated residues and the domain in question where appropriate in the manuscript.

The authors call their tweezers experiments several times single-molecule experiments. I find

this a bit confusing: sure, 1 dsDNA molecule is used, but the binding of many (in fact: how many???) thousands, ten thousands) molecules of A3G is measured, not single binding events.

Response: The phrase single molecule is in fact referring to the single DNA construct. While there are many A3G binding events observed, we use the phrase to differentiate from bulk assays (such as results seen by running a gel) in which the observed result is the average behavior of many DNA constructs.

p.1 "fast and slow binding state". I found this confusingly colloquial, particular here in abstract. Fast and slow should be between quotes, or explained a bit more precise. What about mentioning it a 2-step binding process, initial reversible binding followed (on a slower time scale) by oligomerization resulting in very stable binding.

Response: The slow and fast designation is a holdover from the previous work before we were fully able to attribute the two binding states to oligomerization. We are now more explicit in referring to the two steps as monomer binding and oligomerization. However, we think it is helpful to use these designations when referring to the earlier work and before deriving the final steps in the manuscript.

p. 6 "As compared to WT A3G, a large..." The (or at least this) reader would be helped with a reference to fig 3B here.

Response: We have added the suggested figure reference.

p. 7 I found the paragraph on top a bit confusing. I guess that the main cause is "The total fractional A3G binding". If I understand it correctly, what the authors mean is at long time or after equilibration. Maybe it would be good to explicitly say this.

Response: Correct, this has been made explicit.

p. 7 "Indeed, as the time-scale for fast monomer association" I found this sentence very complicated and difficult to understand. Please break up and explain with more words, in more detail.

Response: This sentence removed entirely as it was redundant with another sentence.

p. 10 Top paragraph. I found the use of "rapid movement" and "slow dimers" and "slow oligomers" and "slow binding kinetics of dimers" rather confusing. Maybe the authors could be more explicit in referring to a "slow/fast binding equilibrium" and "slow/fast diffusion".

Response: These statements have been rewritten as suggested.

p. 11 One but last sentence of discussion: "()DNA substrate" I guess this should be (-).

Response: Thank you. we have corrected this typo.

fig. 3 please indicate A3G concentration.

Response: We have changed the figure caption to now include the A3G concentration (50 nM) as suggested.

Reviewer #3

Reviewer Comment 1: One of their tools is AFM, where they show that the wildtype forms a sequence of oligomers on SS DNA while several mutants exhibit reduced ability to oligomerize. Unfortunately, there are no images from such experiments and the information as to how the volumes of the bound proteins were measured is limited (Gwyddion segmentation function is not an adequate description of method). Such data and information should at least be part of supplemental materials.

Response: We have now included a supplemental figure showing representative AFM images of a strand of linearized m13 ssDNA with different forms of A3G bound. The exact procedure for determining protein cluster size is also now described in the manuscript methods.

Reviewer Comment 2: In addition, wouldn't it be possible to examine the binding of the enzymes as a function of concentration and time under the AFM to corroborate the results of the tweezers? Events may be too fast for AFM imaging but they can probably still be captured with the help of substrate binding that may, in principle, slow dissociation significantly.

Response: The purpose of the AFM results was to confirm the oligomerization deficiency of the specific mutants used in this study, as predicted by A3G structure. AFM imaging has many limitations that prevent us from fully comparing these results with those from our optical tweezers. For example, ssDNA is very flexible and tends to fold over on itself making precise imaging difficult. In order to distinguish individual A3G clusters, we must observe at lower A3G concentrations (or specifically a small ratio of A3G protein to DNA). The saturating conditions we observe using optical tweezers would result in large aggregates, which cannot be individually resolved. It is possible to overcome some of these challenges with specific techniques and binding constructs (see work of the Lyubchenko group referenced in the manuscript), but this remains outside the scope of this project.

REVIEWERS' COMMENTS:

Reviewer #1 (Remarks to the Author):

The authors have addressed my concerns, and I think those of the other referee. The manuscript is now suitable for publication.

Reviewer #2 (Remarks to the Author):

The authors have carefully addressed my and the other reviewers' comments and improved the manuscript. A key aspect is that the relationship with previous work is now much clearer. This makes an assessment of the novelty aspect of the current study more obvious: this represents an important step forward in our understanding of apobec proteins. I support publication in Nature Communication in current form

Reviewer #3 (Remarks to the Author):

This reviewer appreciates the addition of the AFM figures and the better description of the procedure by which the bound stoichiometries of the bound A3G and its mutants are estimated.

In principle, my concerns from the first submission have mostly been addressed and I recommend acceptance of the manuscript.

However, it would be good to add a reference image of the ssDNA alone to clearly contrast with the images of bound enzymes. It is known that ssDNA folds in all ways making AFM imaging challenging, making the choice of a threshold in order to measure volumes of small bound molecules in a consistent manner, particularly so.

The enzyme concentrations used for AFM (70nM) are in the mid-range of what was used in the tweezer experiments (20-200nM). The degree of enzyme saturation is difficult to discern in the images provided, so a reference (control) ssDNA image (and maybe a sample thresholded one with enzyme) would be helpful.

Response to reviewers

Reviewer #1 (Remarks to the Author):

The authors have addressed my concerns, and I think those of the other referee. The manuscript is now suitable for publication.

Author Response: We thank the reviewer for their helpful comments in revising the manuscript.

Reviewer #2 (Remarks to the Author):

The authors have carefully addressed my and the other reviewers' comments and improved the manuscript. A key aspect is that the relationship with previous work is now much clearer. This makes an assessment of the novelty aspect of the current study more obvious: this represents an important step forward in our understanding of apobec proteins. I support publication in Nature Communication in current form

Author Response: We thank the reviewer for their constructive comments.

Reviewer #3 (Remarks to the Author):

This reviewer appreciates the addition of the AFM figures and the better description of the procedure by which the bound stoichiometries of the bound A3G and its mutants are estimated. In principle, my concerns from the first submission have mostly been addressed and I recommend acceptance of the manuscript.

However, it would be good to add a reference image of the ssDNA alone to clearly contrast with the images of bound enzymes. It is known that ssDNA folds in all ways making AFM imaging challenging, making the choice of a threshold in order to measure volumes of small bound molecules in a consistent manner, particularly so.

The enzyme concentrations used for AFM (70nM) are in the mid-range of what was used in the tweezer experiments (20-200nM). The degree of enzyme saturation is difficult to discern in the images provided, so a reference (control) ssDNA image (and maybe a sample thresholded one with enzyme) would be helpful.

Author Response:

We thank the reviewer for the positive comments. We did not optimize imaging for ssDNA alone. However, we have included sample thresholded images as requested for examples of both WT A3G and FW A3G. The images show locations that clearly contain bound protein, which can be distinguished from bare DNA. The additional images are now Supplementary Figure 1E and F.